# Effects of Hypoxia on RNA Cargo in Extracellular Vesicles from Human Adipose-Derived Stromal/Stem Cells

**DOI:** 10.3390/ijms23137384

**Published:** 2022-07-02

**Authors:** Benjamin Koch, Alec Geßner, Samira Farmand, Dominik C. Fuhrmann, Andreas G. Chiocchetti, Ralf Schubert, Patrick C. Baer

**Affiliations:** 1Division of Nephrology, Department of Internal Medicine III, University Hospital, Goethe University, 60596 Frankfurt am Main, Germany; b.koch@med.uni-frankfurt.de (B.K.); alec.gessner48@gmail.com (A.G.); samira.farmand@web.de (S.F.); 2Faculty of Medicine, Institute of Biochemistry I, Goethe-University Frankfurt, 60590 Frankfurt am Main, Germany; fuhrmann@med.uni-frankfurt.de; 3Department of Child and Adolescent Psychiatry, Psychosomatics and Psychotherapy, University Hospital, Goethe University, 60528 Frankfurt am Main, Germany; andreas.chiocchetti@kgu.de; 4Division of Allergology, Department for Children and Adolescents, Pneumology and Cystic Fibrosis, University Hospital, Goethe University, 60590 Frankfurt am Main, Germany; ralf.schubert@kgu.de

**Keywords:** adipose-derived stromal/stem cells, mesenchymal stromal/stem cells, hypoxia, extracellular vesicles, translatomics, RNA, mRNA, microRNA

## Abstract

Mesenchymal stromal/stem cells and their derivates are the most promising cell source for cell therapies in regenerative medicine. The application of extracellular vesicles (EVs) as cell-free therapeuticals requires particles with a maximum regenerative capability to enhance tissue and organ regeneration. The cargo of mRNA and microRNA (miR) in EVs after hypoxic preconditioning has not been extensively investigated. Therefore, the aim of our study was the characterization of mRNA and the miR loading of EVs. We further investigated the effects of the isolated EVs on renal tubular epithelial cells in vitro. We found 3131 transcripts to be significantly regulated upon hypoxia. Only 15 of these were downregulated, but 3116 were up-regulated. In addition, we found 190 small RNAs, 169 of these were miRs and 21 were piwi-interacting RNAs (piR). However, only 18 of the small RNAs were significantly altered, seven were miRs and 11 were piRs. Interestingly, all seven miRs were down-regulated after hypoxic pretreatment, whereas all 11 piRs were up-regulated. Gene ontology term enrichment and miR-target enrichment analysis of the mRNAs and miR were also performed in order to study the biological background. Finally, the therapeutic effect of EVs on human renal tubular epithelial cells was shown by the increased expression of three anti-inflammatory molecules after incubation with EVs from hypoxic pretreatment. In summary, our study demonstrates the altered mRNA and miR load in EVs after hypoxic preconditioning, and their anti-inflammatory effect on epithelial cells.

## 1. Introduction

Extracellular vesicles (EVs) provide essential intercellular communication and are associated with a variety of different physiological processes in healthy tissues as well as in pathophysiological contexts such as tissue regeneration [1,2,3]. Based on their biogenesis and size, EVs can be divided into exosomes and microvesicles. Depending on the cell of origin and its microenvironment, the released EVs are different and individual [4], which is reflected in particular in the composition of their surface molecules and their loading. EVs are detectable in almost any body fluid, and are described to transfer their intercellular information to neighboring or distant cells in the form of proteins, lipids, and different RNAs. In this context, EVs bring bioactive components from donor to recipient cells. The molecules introduced in this way regulate gene expression and alter cellular functions in the recipients. It is generally accepted that the main therapeutic effects come from the mRNA and miR cargo, but proteins and lipids also play a role. A landmark in EV research was the demonstration that the cargo of EVs contains both mRNA and miR and that EV-associated mRNAs can be translated into proteins by target cells [5,6].

The beneficial effects of mesenchymal stromal/stem cells (MSCs) applications in regeneration are also attributable to paracrine signaling, which includes the release of EVs [7,8]. Transplantation of MSCs or their conditioned medium including EVs requires cells with maximal regenerative capacity [9]. In the last decade, optimization of beneficial effects of cell therapy has been studied to improve transplantation and paracrine properties of cell therapeutics. Recent data suggest that the regenerative potential of stromal/stem cells can be enhanced by in vitro preconditioning using environmental or pharmacological stimuli, thereby significantly increasing their therapeutic efficacy [4,10,11]. Cellular responses to different preconditioning methods are complex, as they either trigger or suppress different molecular signal transduction cascades. Moreover, the preconditioning procedure affects a variety of factors rather than a single, specific RNA or miR.

The aim of the present study was the isolation and characterization of EVs from hypoxic preconditioned adipose-derived stromal/stem cells (ASCs). For this purpose, EVs were isolated from ASCs cultured under the normoxic and hypoxic environment by size exclusion chromatography and characterized. The RNA loading of the EVs was then characterized by RNA- and miR-sequencing techniques. In addition, the effect of EVs on the expression of anti- and pro-inflammatory molecules in renal tubular epithelial cells was investigated.

## 2. Results

### 2.1. Isolation and Characterization of EVs

For EV isolation, ASCs were grown to subconfluency (approximately 11,000 cells/cm^2^) under standard cell culture conditions. The cells were washed before the serum-containing culture medium was replaced with pure medium lacking FBS in order to assure that all EVs originated from ASCs in the following isolation process. Afterward, cells were cultured for 48 h either under standard (normoxic, 21% O_2_) or under hypoxic (1% O_2_) conditions as a preconditioning regimen. ASCs retained their characteristic spindle-shaped fibroblastic morphology, even in serum-free culture, both under normoxic and hypoxic conditions for 48 h (data not shown). The conditioned medium was collected and used for further experiments and EV isolation via size exclusion chromatography (SEC). In total, media samples were collected from ten different ASC isolations (biological replicates).

Isolated EVs were examined for their expression of characteristic surface molecules CD9, CD63, and CD81 by flow cytometry. Capture antibodies targeting CD9, CD63 and CD81, which were coupled to magnetic Dynabeads were used to bind EVs expressing these specific molecules. The expression levels were then quantified with fluorophore-coupled detection antibodies. To define the gating strategy, a reference measurement was carried out with 2.8 μm Dynabeads M-280 (Figure 1B) by plotting the forward scatter (FSC-A) against the side scatter (SSC-A) (Figure 1A–C, red circles). PBS was used as a control and showed a background signal which was eliminated by the gating strategy (Figure 1A). The Dynabead-coupled EVs were then analyzed. The result of a representative measurement is depicted as a histogram plotting the fluorescence intensity of the detection antibody versus the number of detected events. A signal shift compared to the isotype control represented a positive detection of the respective marker. Flow cytometric analysis revealed that ASC-derived EVs were positive for CD9, CD63 and CD81 (Figure 1D–F).

In addition, isolated EVs were characterized by nanoparticle tracking analyses (NTA) (Figure 2B,C). For calibration of the NTA measurements we used 100 nm commercially available polystyrene microspheres (Figure 2A) and distilled water (data not shown). Representative NTA measurements of EVs isolated from ASCs after culture in a normoxic (nEV) or hypoxic environment (hEV) for 48 h are shown in Figure 2B,C. The measured average particle size of seven measurements was calculated. The size of isolated EVs ranged between 43 and 420 nm with a mean of 167.0 ± 14 nm in nEV isolations, and a mean of 179.7 ± 9 nm in hEV isolations. In the measurements, no particles bigger than 420 nm were detected. For visualization, isolated EVs were stained with the fluorescent dye PKH26, fixed on an adhesion slide, and detected by fluorescence microscopy. (Figure 2E).

### 2.2. Whole RNA Transcriptomics

Following EV isolation from CM of ASCs which were cultured under normoxic or hypoxic conditions, total RNA cargo in isolated EVs was examined. Therefore, RNA was extracted using a Total Exosome RNA and Protein Isolation Kit, and RNA concentration was measured with a NanoDrop spectrophotometer (NanoDrop). The mean concentration of RNA in hEVs (15.8 ng/µL) was lower than in nEVs (24.4 ng/µL). A maximum RNA concentration of 84.7 ng/µL was obtained from EVs under normoxic conditions, while the highest RNA concentration in hypoxic preconditioned EVs was 60.7 ng/µL. Although RNA concentration was low, it was sufficient to perform RNA sequencing using NEBNext Single Cell/Low Input RNA Library Prep Kit for Illumina. Therefore, 8 µL of the EV-RNA were used as input for library preparation in duplicates. Afterward, the generated and amplified DNA libraries were sequenced using an Illumina NextSeq 550 (single-end, 75 cycles). Bioinformatic processing of the raw data was performed using kallisto and the Ensembl coding RNA database [12,13].

A differential expression analysis for mRNA was performed using DESeq2 in iDEP.951 [14]. No significant differences in total read counts were detected after normalisation. For data analysis, sequences were aligned to known mRNAs. In total, 19,0069 transcripts were detected in four samples, and 57,372 of these passed the filter settings used in iDEP.951 (minimum 0.5 counts per million). Transcripts with an FDR lower than 0.05 and a log2-fold change ≥2 or ≤−2 were considered differentially expressed. From the detected hits, we found 3131 to be significantly regulated (Appendix A). Data are shown as an MA plot (M (log ratio) versus A (mean average) scales (Figure 3A), and a heatmap (Figure 3B) showing differentially expressed transcripts.

It was confirmed that hypoxia results in EV transcriptome changes. Interestingly, only 15 transcripts were found to be down-regulated, while 3116 transcripts were up-regulated after hypoxic preconditioning (Appendix A). In addition, selected statistically significant increased and decreased transcripts are shown in Table 1.

Next, a gene ontology (GO) term enrichment analysis was performed in order to study the biological background of the detected RNAs. Therefore, the bioinformatic analysis tools PANTHER [15] and shinyGO [16] were used to test for enrichment among biological and cellular processes (Figure 4). The most significantly enriched cellular processes are shown in Figure 4A. Cytoskeletal regulation and the ubiquitin proteasome pathway stand out very prominently, with more than five genes being affected by hypoxia, while ATP and serine glycine biosynthesis, as well as glycolysis, were most strongly upregulated. The GO terms for biological processes are displayed in Figure 4B. The pies represent the percentage of differentially expressed RNAs listed with the respective GO term. Most RNAs where associated with the GO terms “cellular process”, followed by “metabolic process” and “biological regulation” (Figure 4B).

### 2.3. Small RNA Transcriptomics

For further characterization of EVs, small RNA loading was characterized by next generation sequencing (NGS) on an Illumina MiSeq. After NGS analysis, no significant differences in total read counts were detected after normalisation. 192 small RNAs were detected, 190 passed the filter settings used in iDEP.951 (Appendix A), and of these 169 were miRs and 21 were piwi-interacting RNAs (piRs). To illustrate the up- and down-regulated small RNAs, two clusters of selected down- (Figure 5A) and up-regulated small RNAs (Figure 5B) are shown by heatmaps (significant and non-significant).

However, of these detected small RNAs, only 18 were significantly altered between hEVs and nEVs (FDR < 0.1 and fold change ≥2 or ≤−2 were considered differentially expressed) were considered differentially expressed). Of these, seven were miRs and 11 were piRs. Interestingly, all seven significantly regulated miRs were down-regulated after hypoxic pretreatment, whereas all 11 piRs were up-regulated (Table 2).

Finally, the network of the seven significantly regulated miRs (see Table 2) and potentially interacting genes was visualized using miRNet2.0 [17]. The network visualization and the boxplot graphs of the seven miRs are shown in Figure 6A,B. The functional properties of the identified miRs were then characterized in more detail. For this purpose, miR Reactome and GO analysis of biological processes (GO-BP) were analyzed using the functional explorer in miRNet2.0 [17]. The main hits for Reactome analysis were “cellular response” (196 hits), “immune system” (147), “disease” (129), and “metabolism of proteins” (123). The main hits for biological processes were “regulation of protein metabolism” (264 hits), “regulation of biological pathways” (244), “nuclear transport” (233), “regulation of protein modification” (190), and “regulation of cell cycle” (144).

Furthermore, miR-target enrichment using miRTarBase, a curated database of microRNA-target interactions [18], showed that a huge amount of the detected miR interact with mRNAs found to be regulated upon hypoxia (Figure 6C,D).

For example, cathepsin D (CTSD, see Table 1) interacts with miR-26b-5p (and four other nonsignificant miRs detected in EVs). A total of 31 miRs were detected that interact with BCL2 (Figure 6C), as well as a variety of BCL-like genes, including the significantly regulated miR-125b-5p. Twenty-seven miRs interact with TGFBR3, including two significantly regulated let-7 miR (let7a-5p and let7e-5p). The analysis also showed that LIN28A interacts with four significantly down-regulated RNAs (Figure 6D and Table 3).

Network analysis showed the described interaction of the significantly down-regulated miRs with several mRNAs coding for regeneration-promoting and anti-inflammatory proteins (Table 3). For example, mRNAs coding for IGF2 were described to interact with miR-125b-5p and let-7a-5p. Transcripts of some of the interacting genes were found to be up-regulated in hEVs after hypoxic pretreatment (highlighted in red). Nevertheless, many other transcripts of the genes shown in Table 3, which interact with one of the seven miRs, were also found in ASC-EVs but not regulated between nEV and hEV.

### 2.4. Effect of EVs on Renal Epithelial Cells

Finally, after confirming the uptake of ASC-EVs in cultured renal epithelial cells (Figure 7A), the therapeutic effect of the preconditioned EVs on subconfluent human primary renal tubular epithelial cells (TECs) should be verified. For this, injured subconfluent TECs were stimulated with EVs from normoxia or hypoxia (8 µg EV-protein per 10^5^ cells) and gene expression of HGF, IDO-1, IL-10, IL-6, and TNFα was determined (from four biological replicates). Relative gene expression to the controls (medium without EVs) was expressed as x-fold expression using the ΔΔCT method.

While there was no change in the expression of the pro-inflammatory cytokines IL-6 (Figure 7C) and TNFα (Figure 7B), all three anti-inflammatory molecules tested showed an increase in their expression (Figure 7D–F). Interestingly, the induced expression was only significant after incubation with hEVs. Incubation with nEVs also induced expression of the three anti-inflammatory markers but without reaching significance. Nevertheless, a clear expression enhancing trend could be demonstrated.

## 3. Discussion

Efforts to elucidate the precise mechanisms of stem cell action in regenerative medicine therapies have highlighted the prominent role of paracrine activity in promoting regeneration rather than cell engraftment and differentiation. In this context, extracellular vesicles have been investigated extensively in recent years. The cargo of extracellular vesicles plays a major role in cell-cell communication. Due to the ability of EVs to transport large cargos of nucleic acids, lipids, and proteins, which can alter the function of single or multiple target cells, they have been considered as a vehicle of active biomolecules in order to reduce inflammation or support tissue repair [19]. In this context, the release of EVs from mesenchymal stromal/stem cells seems to be one major mechanism to enhance regeneration and immune modulation [20]. It is also described that the secretome of MSCs can be modified by in vitro preconditioning regimens such as hypoxia. The pretreatment also seems to alter the cargo of released EVs and thereby enhance their regeneration-promoting, immunomodulatory, and anti-apoptotic potential [20,21]. Currently, the EV cargo after pretreatment regimes, like hypoxia, is not comprehensively characterized. Nevertheless, it has been shown that EVs from MSCs are loaded with a heterogeneous group of coding and non-coding RNA-molecules [22]. Furthermore, it has been revealed that EVs predominantly contain miRs with a size < 300 nt as well as intact and fragments of mRNA and long non-coding RNA. However, compared to the respective content in the originating cell, the RNA concentration of EVs is lower [23].

With regard to the isolation method of EVs, different procedures have been described in recent years that influence the EV composition and make it difficult to compare the published results from the various isolation methods (e.g., differential ultracentrifugation, size-exclusion chromatography and several commercial kits). In this work, EVs have been isolated based on size difference via SEC using sepharose columns [24], where filtering and concentration steps were necessary in order to decrease the culture medium volume and to remove cell contaminations and apoptotic bodies. The efficacy and reproducibility of this isolation which also highly preserves EV structure and content has been proven recently [1,24,25]. In this context, however, it should also be mentioned that we filtered the harvested conditioned medium through a 0.45 µm filter and thus apoptotic vesicles and larger EVs were eliminated. Using nanoparticle tracking analysis, we were able to show here and in an older study [26] that no change in either EV size and protein content nor in concentration occurred after hypoxic preconditioning.

The culture of ASCs under hypoxic conditions has been shown to increase the transcription and trafficking of several molecules that are involved in regenerative, anti-apoptotic, pro-proliferative and anti-inflammatory processes. A differential expression analysis for mRNA showed that 3131 coding RNAs were significantly regulated upon hypoxic pretreatment. Interestingly, only 15 of these hits were downregulated, but 3116 were upregulated. In addition, miR content of nEVs and hEVs was analyzed by NGS. However, in contrast to the mRNA data, all seven significantly regulated miRs were down-regulated after hypoxic pretreatment, and only the detected significant 11 piRs were up-regulated.

Hypoxia leads to the stabilization of the transcription factor hypoxia-inducible factor-1α (HIF-1α). The factor is then translocated to the cell nucleus, which results in the expression of proangiogenic genes, e.g., VEGF [4,27]. In this context, our data show that the mRNA of an inhibitor of HIF-1α (HIF1AN) was downregulated under hypoxia (−4.8x, not significant, data not shown). The molecule functions as an oxygen sensor and its hydroxylation prevents the interaction of HIF-1α with transcriptional coactivators under normoxic conditions and is involved in transcriptional repression through interaction with HIF1α and histone deacetylases [28]. Furthermore, cathepsin D (CTSD) has been shown to activate the growth factor VEGF [29]. CTSD mRNA is highly upregulated in our study (29.37-fold, Table 1), but five of the CTSD-interacting miRs (e.g., miR-26b-5p) were down-regulated, which in addition to induction by HIF might explain VEGF mRNA upregulation after hypoxia. Hypoxia has also been shown to induce insulin-like growth factor 2 (IGF2), a major growth factor and anti-apoptotic signaling molecule. In our previous study, the mRNA-expression of VEGF and IGF2 has been shown to be enhanced as a cellular response to hypoxia [26]. Here, the significant induction of IGFBP4 (10.62-fold) and 6 (27.55-fold) were shown. Consistent with this, IGFBP4 released by MSCs has been shown to be one of the major survival factors secreted by MSCs [30]. Among the transcripts involved in apoptotic processes are the transmembrane BAX inhibitor motif containing 4 (TMBIM4), cofilin 1 (CFL1), prostaglandin E2 synthase (PTGES), as well as several BCL2 interacting molecules like BNIP2 and BCL2L12 (Table 1). The induction of CFL1 is in line with an older study from our group. A significant induction of the CFL1 protein has been detected after the preconditioning regimen [26], and CFL1 is further listed at rank 29 of the most detected proteins in EVs. In addition, the transcript of PTGES was significantly up-regulated. It has recently been demonstrated that MSCs secrete PTGES and miR210 upon cultivation under hypoxic conditions, decreasing hepatic oxidative stress and fibrosis in mice with induced liver cirrhosis [31]. Prostaglandin E2 has been shown to be involved in renal tubular epithelial regeneration through the inhibition of apoptosis and epithelial-mesenchymal transition [32]. In this context, EVs from MSCs were shown to protect renal tubular epithelial cells from apoptosis via transfer and subsequent transcription of anti-apoptotic RNAs in vivo [33]. Furthermore, numerous hypoxia-induced transcripts that were involved in immune processes have been identified, namely chemokine ligands (CCL) 2 and 18, heat shock protein 90B1 (HSP90B1), indoleamine-pyrrole 2,3-dioxygenase-1 (IDO1) and metallomatrixprotease 2 (MMP2). Whereas CCL18 was up-regulated, CCL2 was found to be down-regulated in our study. CCL18 has been shown to play a part in both activation of the immune system and immunosuppressive effects by induction of immature dendritic cells and macrophages to differentiate into an immunosuppressive cell type. HSP90B1 has been implicated as an essential immune chaperone to regulate both innate and adaptive immune responses. HSP90 has also been shown to improve MSC viability and protects MSCs against apoptosis induced by hypoxia [34]. Nevertheless, the down-regulation of the IDO1 mRNA load in hEVs is not in line with the up-regulation of anti-inflammatory molecules. The anti-inflammatory molecule IDO1 is involved in the suppression of potentially dangerous inflammatory processes in the body. Another significantly down-regulated hit is the mRNA of the nucleosome assembly protein 1-like4 (NAP1L4). NAP1L4 has been shown to regulate cell fate by controlling the expression of p53-responsive pro-arrest and pro-apoptotic genes [35].

MicroRNAs are small non-coding RNA molecules responsible for RNA silencing and post-transcriptional regulation of gene expression by binding to mRNA, leading to mRNA degradation or inhibition of protein synthesis. In addition to the mRNA cargo to EVs, miR composition plays an important role in their biological function, such as cell cycle regulation, apoptosis, migration, inflammation, and angiogenesis, among others [36]. EVs from hypoxic pretreated ASCs showed decreased levels of let-7 family miRs, which are responsible for development and tumor suppressor functions [37]. Present studies suggested that let-7 targets the mRNA of IGF-binding proteins (IGF-BP) and IGF. The mRNAs of several IGF-BP (−4, −6) were shown to be significantly upregulated in our study (Appendix A), the upregulation of IGF2-mRNA in response to hypoxia has been shown in an older study as well [26]. Moreover, in our study, hEVs were less loaded with the anti-apoptotic miR-125b-5p than nEVs. In contrast, others have demonstrated a significant enrichment of miR-125b-5p in hEV compared to nEV using an miR array [38]. The administration of miR-125b-knockdown-EVs significantly increased the infarction area and suppressed cardiomyocyte survival in an in vivo model of myocardial infarction [38]. Mechanistically, miR-125b-knockdown-EV lost the capability to suppress the expression of proapoptotic genes. Other have shown that EVs from MSCs significantly attenuated cell cycle arrest and apoptosis in renal epithelial cells in vivo and in vitro, and that inhibition of miR-125b-5p mitigated the protective effects of MSC-EVs [39].

miR-target enrichment using miRTarBase, a curated database of microRNA-target interactions [18], showed that a huge amount of the detected miR (significant and not significantly regulated) interact with mRNAs found to be regulated upon hypoxia (Figure 6C,D). For example, miR-target enrichment analysis has shown that four of the down-regulated miR interact with LIN28A (Figure 6D), and, while not significantly, LIN28A and B mRNAs are also both downregulated after hypoxia in our data (−7.51-fold respectively −3.27-fold). LIN28A and B have also been shown to encode RNA-binding proteins that enhance the translation of the IGF-2 mRNA [40].

Finally, we investigated the effects of hEVs and nEVs on injured renal epithelial cells. Using this well characterized in vitro model, the effects of EVs on the expression of pro- and anti-inflammatory molecules was investigated. Therefore, subconfluent renal tubular epithelial cells were used, as they share several characteristics with wounded epithelial cells in the regeneration process [41]. TECs used in the study were shown to express vimentin, which clearly indicates their injured character [41]. The co-expression of cytokeratin and vimentin in cultured epithelial cells indicates the immature or wounded character of the cells [42]. The intermediate filament protein vimentin is normally expressed in mesenchymal cells and is not present in differentiated epithelial cells. We found no change in the expression of the two pro-inflammatory cytokines, but all three anti-inflammatory molecules significantly increased their expression after incubation with preconditioned EVs. Others have shown that MSCs protect renal tubular epithelial cells from apoptosis via EV-mediated transfer and the subsequent transcription of anti-apoptotic RNA in vivo [33]. We also detected a slight downregulation of BCL-2 and several BCL-like RNAs, although this was not significant (data not shown). In addition, 31 miRs were detected to interact with BCL2 (Figure 6C), as well as a variety of BCL-like genes, including the significantly regulated miR-125b-5p. It was further shown that depletion of miRs in MSCs decreased the renal regenerative properties of these cells and their derived EVs [43]. It was found that the EV-mediated transfer of miRs to damaged tubule epithelial cells resulted in functional recovery [44], whereas treatment of tubular epithelial cells with miR-knockout EVs did not improve molecular injury [43]. In conclusion, miR depletion in EVs from MSCs significantly reduced their intrinsic regenerative potential, suggesting a critical role of miRs in epithelial cell recovery. In regeneration, the protective effects of MSCs are not attributed to their differentiation into epithelial or endothelial cells, but to enhanced regulation of anti-inflammatory and organ-protective mediators (such as IL-10, HGF, and different other molecules like miRs).

One limitation of the current study is the number of groups in the mRNA and miR sequencing, as an increase would improve the analysis of the data collected (currently only two versus two in both sequencing). Furthermore, the analysis of EV-RNA and miR remains difficult due to very low RNA and miR-concentrations. Further analyses for more efficient EV-RNA and miR isolation methods from EVs are required. Alternately, the upscaling of cell number and/or volume of culture medium which is used for EV isolation can increase EV yields and consequently RNA and miR concentrations. Further work is also required to validate selected miR hits by PCR analysis and to investigate causal mechanisms by knockdown of individual miR.

The aim of the present study was the isolation and characterization of EVs from hypoxic preconditioned adipose-derived stromal/stem cells (ASCs). For this purpose, EVs were isolated from ASCs cultured under the normoxic and hypoxic environment by size exclusion chromatography and characterized. The RNA loading of the EVs was then characterized by RNA- and miR-sequencing techniques. In addition, the effect of EVs on the expression of anti- and pro-inflammatory molecules in renal tubular epithelial cells was investigated. It was clearly shown that this in vitro pretreatment significantly affected ASC EVs, resulting in an altered EV mRNA and miR cargo. The in vitro assay clearly demonstrated the induction of anti-inflammatory mRNA in renal tubular epithelial cells. The use of EVs as paracrine effectors in cell-free, regeneration-promoting therapies may reduce the risk of side effects, but this approach requires vesicles with maximal regenerative capacity. The cultivation of ASCs under hypoxic conditions has shown promise as an in vitro preconditioning method that enhances the anti-oxidant, anti-inflammatory, and regenerative effects of EVs. These properties may provide new potential therapeutic options for the treatment of diseases for which there are limited, mostly supportive, treatment options. In summary, EVs isolated from hypoxically preconditioned cells represent a particularly promising approach for regenerative medicine.

## 4. Materials and Methods

### 4.1. Cell Isolation, Culture, and Characterization

Human adipose-derived stromal/stem cells (ASCs) were isolated from adipose tissue obtained from nine female donors undergoing cosmetic liposuction. Aspirated tissue was digested at 37 °C with 0.075% collagenase I (CellSystems, Troisdorf, Germany) under continuous agitation for 45–60 min. The stromal–vascular fraction was separated from the remaining fibrous material and the floating adipocytes by centrifugation at 300× *g*. The sedimented cells were washed with phosphate-buffered saline (PBS) and filtered through a 100 µm pore filter (Millipore, Schwalbach, Germany). Erythrocyte contamination was reduced by density gradient centrifugation (Bicoll; Biochrom, Berlin, Germany). Cells were then plated for initial cell culture and cultured at 37 °C in an atmosphere of 5% CO_2_ in humid air (Normoxia). Primary cell isolates and cultured cells were fully characterized as described previously [45,46]. Dulbecco’s Modified Eagle’s Medium (DMEM; Sigma, Taufkirchen, Germany) was used with a physiologic glucose concentration (100 mg/dL) supplemented with 10% fetal bovine serum (FBS; No. S0615, Lot 0001640839, Sigma/Merck, Darmstadt, Germany) as the culture medium. The medium was replaced every three days. Cells were passaged at 85–90% confluency by trypsinization. ASC in passage 1–4 were used for the experiments. The cell morphology was examined by phase contrast microscopy, as shown previously [26]. Expression of characteristic ASC markers (CD73, CD90, and CD105) and tri-lineage differentiation of cultured ASCs was proven, as described previously [26,47].

### 4.2. Preconditioning of ASCs

ASCs were either cultured under standard conditions (controls in normoxia, 21% O_2_) or preconditioned by incubation in a hypoxic environment (1% O_2_) [26]. For this purpose, cells were grown to subconfluency and were washed twice with PBS. Cells received fresh serum-free low-glucose DMEM without supplements and were then placed in an InvivO_2_ 400 at 1% oxygen for 48 h. Normoxic controls were also cultured in serum-free low-glucose DMEM without supplements and were placed in a Hera cell incubator at 21% oxygen. After 48 h of preconditioning, the cells and the preconditioned medium were collected.

### 4.3. Isolation of Extracellular Vesicles

The preconditioned medium (PCM) was used to isolate EVs from normoxic (nEVs) or hypoxic (hEVs) pretreatment for 48 h (growth area 150 cm^2^ with 16 mL serum-free DMEM). After 48 h incubation, the PCM was centrifuged for 10 min at 600× *g* in order to remove cell debris. The PCM was then filtered using a 0.45 µm PVDF filter and concentrated 10-fold by centrifugation at 1500× *g* for 20 min using 30 kDa molecular weight cut-off Centriprep 30 K filters. The PCM was then either stored at 4 °C for further analysis or used for isolation of EVs. They were isolated from PCM by size exclusion chromatography (SEC) using Sepharose CL-2B columns [1,24]. The PCM was applied onto a Sepharose CL-2B column, which was washed and equilibrated with PBS. As elution buffer, PBS was applied to the column until 18 flow-through fractions with a respective volume of 500 µL were collected. The fractions 7–12 containing extracellular vesicles were pooled and subsequently concentrated (to approximately 400 µL EV solution) using a 3 kDa molecular weight cut-off Amicon filter by centrifuging for 25 min at 2800× *g*. EV samples were either used immediately or stored at −80 °C for further characterization. For selected experiments, isolated EVs were stained with a commercially available PKH26 Red Fluorescent Labeling (Merck, Darmstadt, Germany), as described previously [26].

### 4.4. Characterization of Extracellular Vesicles

To determine the size distribution and concentration of the isolated EVs, we used nanoparticle tracking analysis (NTA) [48]. Therefore, 10 µL of concentrated EV solution (nEVs and hEVs) was diluted 1:100 with nuclease-free water and immediately analyzed using a NanoSight NS500 (Malvern Panalytical, Malvern, UK) according to the manufacturer’s instructions. The NTA technology uses illumination with a laser, and the scattered light was then captured by a camera (20-fold microscopic magnification) allowing the calculation of the hydrodynamic diameter of particles with a diameter of 10–1000 nm. Camera settings were fixed with a camera level of 14 and a camera gain of 1.5, and temperature was set to 28 °C. For each sample, six videos with a duration of 30 s were captured and analyzed with the Nano Sight NTA 3.2 software, setting the threshold to 14 and the gain to 1.5. For calibration of the NTA measurements we used 100 nm commercially available polystyrene microspheres (Molecular Probes, No. F13839) and distilled water as a negative control.

To further analyze EVs by flow cytometry, commercially available Flow Detection Reagent Kits using magnetic Dynabeads with a diameter size of 2.7 µm and 4.5 µm were used (Invitrogen, No. 10620D (anti-CD9), No. 10606D (anti-CD63), and No. 10616D (anti-CD81)). In brief, 40 µL magnetic Dynabead solution was added to 1 mL isolation buffer (PBS, 2% FCS, 1 mM EDTA, 0.1% Sodium azide) and placed on a magnetic stand for 2 min. The supernatant was removed, and the beads were washed with 1 mL isolation buffer and re-suspended in 90 µL isolation buffer. To this suspension, 20 µL EV solution in PBS was added and incubated overnight, rotating at 4 °C. The following day, the sample was washed twice with 1 mL isolation buffer before beads were re-suspended in a 300 µL isolation buffer. Then, a 100 µL bead-EV solution was stained with 20 µL fluorescent-labeled detection antibody and incubated under rotation at RT in the dark for 45 min. Afterwards, the solution was washed once with 1 mL isolation buffers, re-suspended in a 300 µL isolation buffer and transferred to a flow cytometry tube. Detection of CD9 and CD81 was performed using a detection antibody conjugated with Phycoerythrin (anti-CD9-PE (ImmunoTools, Friesoythe, Germany) and anti-CD81-PE (BioLegend, San Diego, CA, USA)), whereas CD63 was detected with an allophycocyanin-coupled detection antibody (anti-CD63-APC (ImmunoTools, Friesoythe, Germany)). In addition, we used commercially available Dynabeads M-280 (Fisher Scientific, Schwerte, Germany, No. 11205D) to calibrate forward and sideward scatter in the flow cytometric measurement. As a control for the fluorescent measurements, a 100 µL bead-EV solution without detection antibody was mixed with 200 µL isolation buffer. All samples were measured using a flow cytometer (BD Biosciences, Heidelberg, Germany) and analyzed on the instrument until 10,000 events were detected. 

### 4.5. Isolation of Whole RNA from EVs for Sequencing

For isolation of EV-RNA, 200 µL EV solution in PBS was homogenized by adding 500 µL NucleoZOL reagent and incubating the mixture for 5 min at RT. The solution was then centrifuged at 12,000× *g* and RT for 15 min and the supernatant was transferred to a new tube. In order to precipitate the RNA, 500 µL 100% isopropanol was added and incubated for 10 min at RT before RNA was pelleted at 12,000× *g* for 10 min. The supernatant was discarded, and the pellet was washed twice using 500 µL 75% ethanol followed by centrifugation at 8000× *g* for 3 min. The pellet was then re-suspended in 20–30 µL H_2_O and RNA concentration was measured using a spectrophotometer (NanoDrop, ThermoFisher Scientific, Darmstadt, Germany).

In order to perform RNA sequencing of EVs, cDNA libraries were prepared using NEBNext^®^ Single Cell/Low Input RNA Library Prep Kit for Illumina^®^ according to the manufacturer. The quality and quantity of amplified cDNA were controlled with a Bioanalyzer by running 1 µL of cDNA on a DNA High Sensitivity Chip. After Bioanalyzer analysis, the DNA library was diluted and denatured for the use in the Illumina NextSeq 550 Sequencing System following the protocol of the manufacturer. Sequencing was performed using a NextSeq 550 Sequencer with single-end-read for 75 cycles. Raw data was downloaded from Illumina BaseSpace and the quality was checked with FastQC [49]. In order to analyze sequencing data, the bioinformatic tool kallisto was employed with standard configurations [12], and the Ensembl (Release 98) cDNA databases [13] was used. Differential expression analysis was done by bioinformatics tools iDEP.951 [14]. Gene ontology (GO) analysis was performed using shinyGO 0.76 [16] and PANTHER [15].

### 4.6. Isolation of miR from ASC-EVs for Sequencing

For sequencing, whole microRNA (miR) was isolated from EVs cultured in serum-free medium under normal and hypoxic conditions using the NuceloSpin^®^ miRNA Plasma Kit (Macherey-Nagel) according to the manufacturer’s instructions. The concentration of RNA was analyzed by the HS RNA Kit for TapeStation 4150 (Agilent). The miR library was prepared using the QIAseq miRNA Library Kit according to the manufacturer’s instructions. After library preparation, DNA concentration was analyzed by using a Qubit dsDNA Assay Kit in Qubit 3.0 Fluorometer (ThermoFisher Scientific, Darmstadt, Germany). In addition, the DNA quality was checked using a HS DNA Kit of the Bioanalyzer 2100 (Agilent, Waldbronn, Germany). Next generation sequencing (NGS) was performed using the MiSeq Reagent Kit v3 (Illumina, San Diego, CA, USA), PhiX Sequencing Control v3, and MiSeqTM Desktop Sequencer. For this, samples were pre-diluted (1:5 or 1:10) and prepared for sequencing according to the manufacturer’s instructions. Samples were loaded into the MiSeqTM Desktop Sequencer in a sequencing cassette and sequencing was performed.

The bioinformatics tools iDEP.951 [14], DESeq2shiny [50] and a in-house scripts were used to statistically analyze and display the miR sequencing data. For the in-house scripts, differential expression analysis was performed in R version 4.2.0 (https://cran.r-project.org/ (accessed on 22 April 2022)). Coverage files were converted into raw counts matrices using the Qiagen pipeline. A priori filtering for sparse read counts has been applied. We performed a hierarchical cluster analysis of raw count data based on the Euclidean distance between samples and the Ward algorithm implemented in R using the normalized read counts of the DESeq2 package (function counts (Reads, normalize = T)) miRs. Functional enrichment analyses were performed based on the Reactome and the GO database selection in miRNet2.0 [17]. Target enrichment was done in MIENTURNET (MIcroRNA ENrichment TUrned NETwork) [51] using miRTarBase.

### 4.7. Effect of EVs on Renal Tubular Epithelial Cells

To show the uptake of EVs isolated from ASCs into cultured TEC, EVs were stained with a commercially available PKH26 Red Fluorescent Labeling kit (Merck, Darmstadt, Germany) and incubated with TECs, as previously described [26].

To further investigate the therapeutic effect of EVs on injured (subconfluent) TECs, cells were incubated with EVs from hypoxic ASCs and compared with EVs from cultures under normoxia. EV concentration was measured indirectly by protein quantification with a commercially available assay (BCA1 protein assay, Sigma Taufkirchen, Germany). TECs were seeded on 6-well plates and incubated overnight to subconfluence. The next day, cells were treated with EV solution (8 µg EVs/10^5^ cells in serum-free Medium 199) or control medium (medium 199 without EVs) for 24 h at 37 °C and 5% CO_2_. RNA was then isolated from TEC by a standard protocol using NucleoZOL (Macherey-Nagel, Düren, Germany). After the RNA extraction, cDNA was synthesized for 30 min at 37 °C using 1 µg RNA, 50 µM random hexamers, 1 mM deoxynucleotide-triphosphate-mix, 50 units of reverse transcriptase (Fermentas, St. Leon-Rot, Germany) in 10× PCR buffer, 1 mM β-mercaptoethanol and 5 mM MgCl_2_. A Hot FIREPol EvaGreen Mix Plus was used (Solis Biodyne, Tartu, Estonia) for the master mix; the primer mix and RNAse-free water were added. Quantitative PCR was carried out in 96-well plates using the following conditions: Twelve minutes at 95 °C for enzyme activation, 15 s at 95 °C for denaturation, 20 s at 63 °C for annealing and 30 s at 72 °C for elongation (40 cycles). Finally, a melting curve analysis was conducted. Products were checked by agarose gel electrophoresis in selected experiments. The PCR fragment quantification was realized using the ABI Prism^®^ 7900HT Fast Real-Time PCR System with a Sequence Detection System SDS 2.4.1 (Thermo Fisher Scientific). Relative quantification was estimated by the ∆∆CT method [52] with β-actin as a housekeeper. The level of target gene expression was calculated using 2^−∆∆Ct^. The PCR products in selected experiments were separated by agarose electrophoresis and observed under ultraviolet illumination. Primer pairs were synthesized by Invitrogen (Karlsruhe, Germany) and are listed in the Appendix A.

### 4.8. Statistical Analysis

The statistical analysis of the measured data as well as their graphic representation was performed with the software GraphPad Prism 7.0. The presented data were calculated as mean ± standard deviation (SD). Gaussian distribution of the data was confirmed with a Shapiro–Wilk test. A Kruskal Wallis test with Dunn’s multiple comparisons test was performed to test differences of PCR analysis data. Results with *p* < 0.05 were determined to be significant (* *p* < 0.05).

In order to analyze whole RNA sequencing data, the bioinformatic tools kallisto with standard configurations and the Ensembl (Release 98) cDNA and non-coding RNA databases were used. The bioinformatics tools iDEP.951, shinyGO 0.76, MIENTURNET and in-house scripts were used to statistically analyze and display the RNA and miR sequencing data.

## Figures and Tables

**Figure 1 ijms-23-07384-f001:**
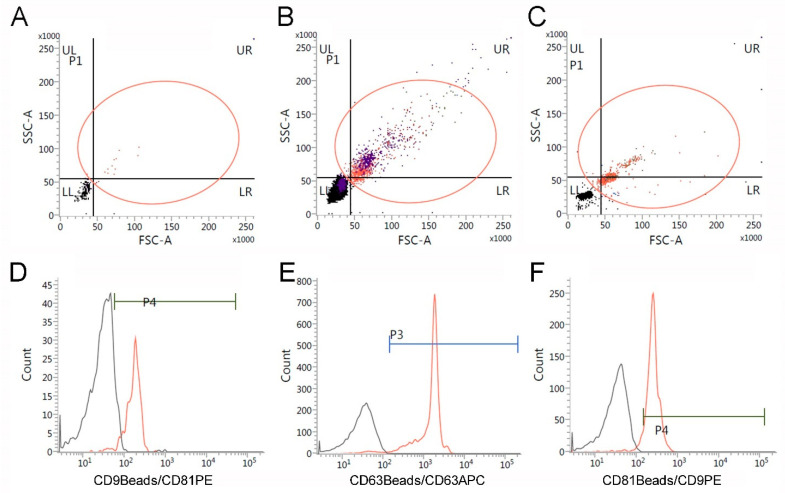
Flow cytometric characterization of extracellular vesicles isolated from adipose-derived mesenchymal stromal/stem cells (ASCs). (**A**) Flow cytometric measurement of PBS as a control with gating strategy. (**B**) Flow cytometric measurement of commercially available Dynabeads M-280 with 2.8 µm diameter to calibrate the measurements. (**C**) Characteristic measurement of the Flow Detection Reagent Kit using magnetic Dynabeads with 2.7 µm diameter. (**D**–**F**) Characterization of EVs with Flow Detection Reagent Kits using conjugated magnetic Dynabeads in combination with conjugated detection antibodies. As controls for the fluorescent measurements, bead-conjugated-EV solutions without detection antibodies were measured. All samples were measured using a flow cytometer (BD Biosciences, Heidelberg, Germany).

**Figure 2 ijms-23-07384-f002:**
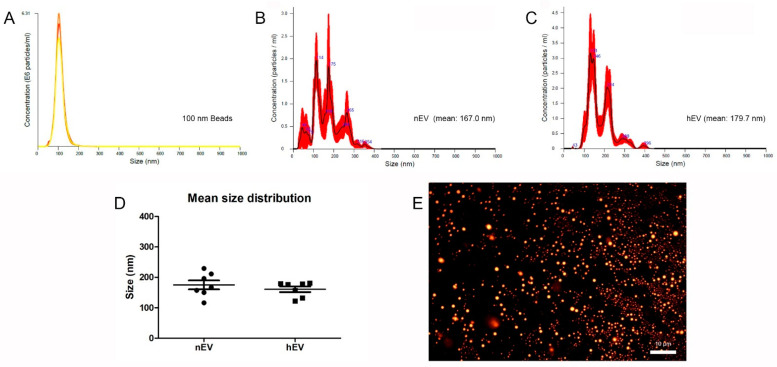
Characterization of extracellular vesicles isolated from adipose-derived mesenchymal stromal/stem cells (ASCs). (**A**) Characteristic nanoparticle tracking analysis of 100 nm Beads. (**B**,**C**) Representative nanoparticle tracking analyses (NTA) of EVs isolated from ASCs after culture in a normoxic (**B**) or hypoxic (**C**) environment for 48 h. (**D**) Calculated mean size distribution of isolated EVs measured by NTA using a NanoSight NS500 (*n* = 7 for nEVs and hEVs). No significant differences were detected. (**E**) PKH26 staining of SEC-isolated EVs. Fluorescence microscopy shows PKH26-labeled EVs on an adhesion slide (scale bar = 10 µm).

**Figure 3 ijms-23-07384-f003:**
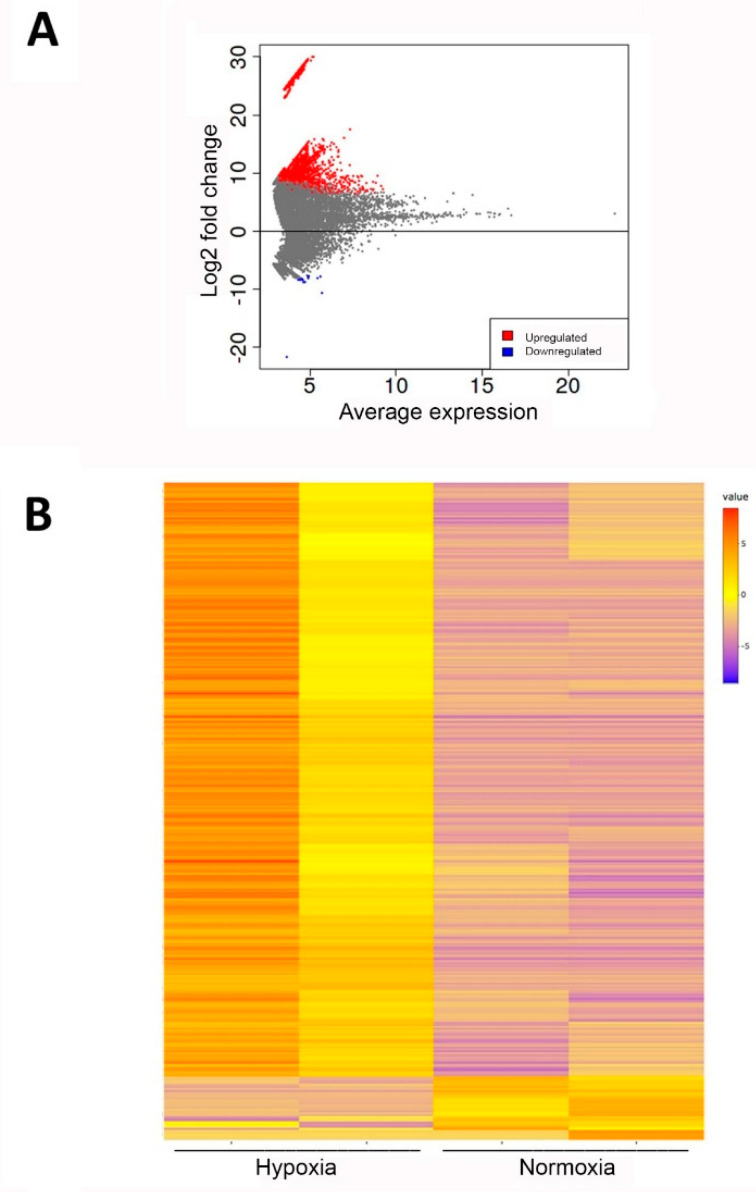
Transcriptome analysis of ASC-EVs after hypoxic versus normoxic culture conditions. (**A**) MA plot of transcript expression changes 48 h after induction of hypoxia, showing significant changes in red (up-regulated) and blue (down-regulated). (**B**) Heatmap of differentially expressed transcripts (blue = significantly down-regulated, red = significantly up-regulated).

**Figure 4 ijms-23-07384-f004:**
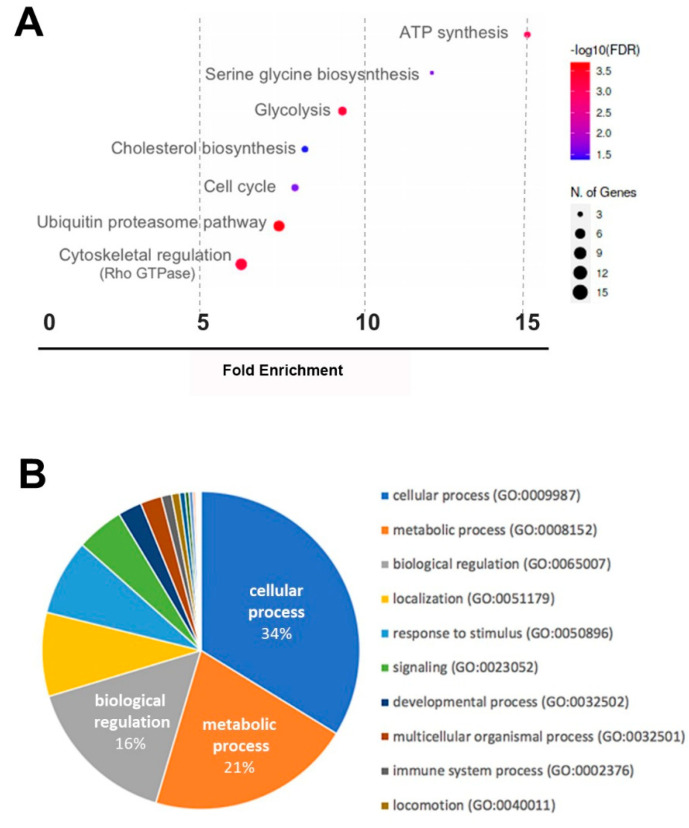
Transcriptome analysis of ASC-EVs after hypoxic versus normoxic culture conditions (significantly regulated transcripts)**.** (**A**) Bar chart based on pathway database “curated. PANTHER” (generated in shinyGO 0.76), (**B**) Functional classification showing biological processes (GO-BP pie chart created with PANTHER v17.0).

**Figure 5 ijms-23-07384-f005:**
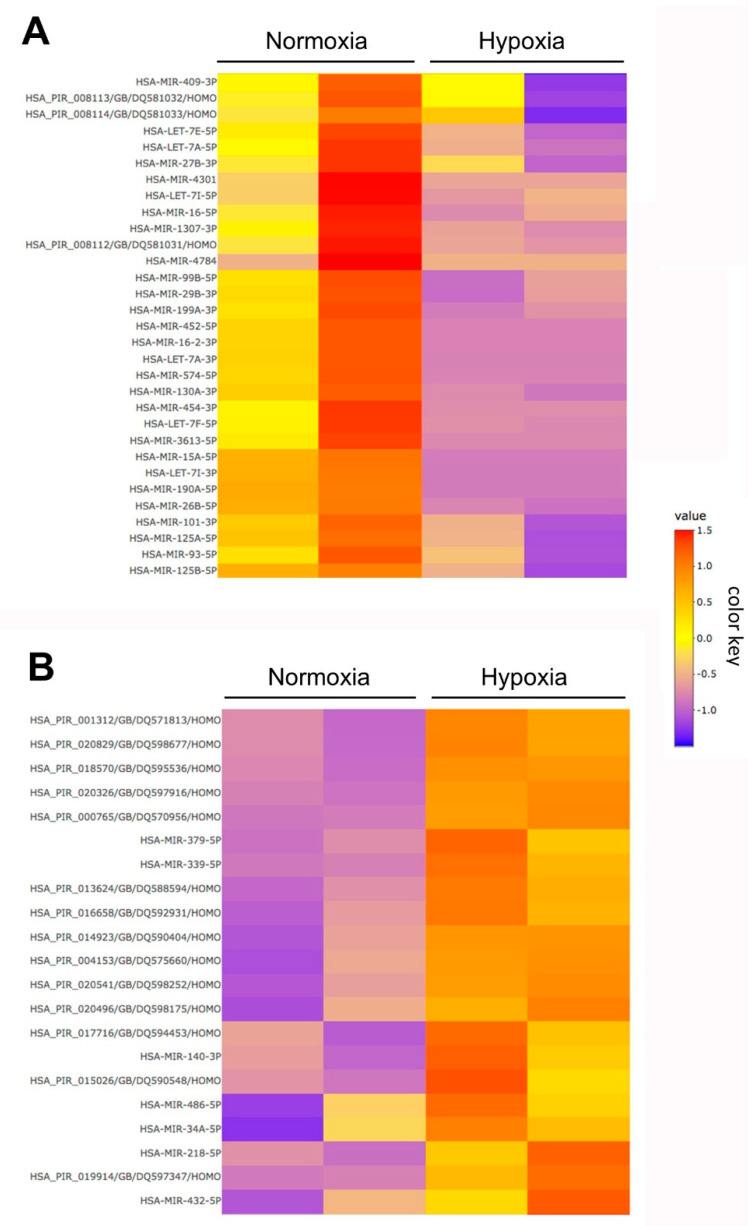
Analysis of miRs expression in ASC-EVs after hypoxic preconditioning. (**A**) Heatmap of selected down-regulated miRs after hypoxic pretreatment (significant and not significant hits). (**B**). Heatmap of selected up-regulated miRs after hypoxic pretreatment (significant and not significant hits).

**Figure 6 ijms-23-07384-f006:**
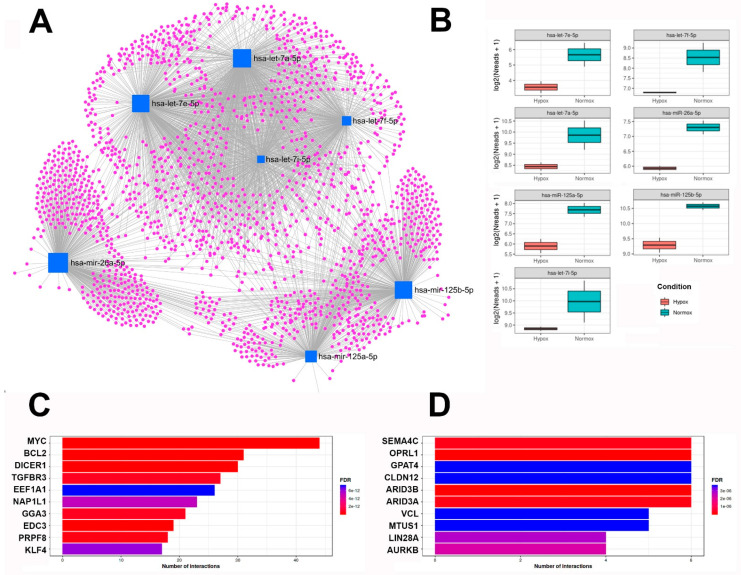
Network visualization and Boxplots of seven significantly regulated miRs. (**A**) Network visualization: miR (blue squares) and the potentially interacting gene-network (magenta dots) created with miRNet2.0. (**B**) Boxplot of regulated miRs (*y*-axis: expression level) created with ggplot2 in R. All seven significantly regulated miRs were downregulated in EVs after hypoxic pretreatment. (**C**) miR-target enrichment with all 169 miR using miRTarBase (in MIENTURNET). Bars show the ten genes (*y* axis: gene symbol) with the highest number of interactions. (**D**) miR-target enrichment using the seven significantly regulated miR (miRTarBase in MIENTURNET). Bars show the ten genes (*y* axis: gene symbol) with the highest number of interactions.

**Figure 7 ijms-23-07384-f007:**
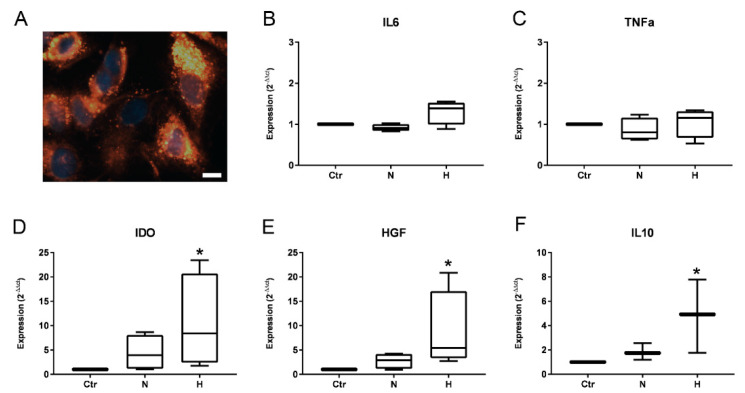
Effect of ASC-EVs on renal epithelial cells in vitro. (**A**) Fluorescence microscopy of PKH26-stained EVs incorporated in renal tubular epithelial cells (blue: staining of cell nuclei with DAPI), bar = 10 µm. (**B**–**F**) PCR analysis. TECs were seeded on 6-well plates and incubated for 24 h (*n* = 4, four biological replicates from four cell isolations). Thereafter, the cell culture medium was replaced with serum-free M199 without EVs (Ctr), with nEVs (N) or with hEVs (H) (8 µg EV protein/10^5^ cells), and incubated for 48 h. Next RNA was isolated using NucleoZol and used for quantitative PCR. Expression of three anti-inflammatory and two pro-inflammatory targets was calculated by 2^−ΔΔct^. Analyses show data from three to four independent experiments (Box-Whisker-plots (Median with Min to Max), *n* = 4 (**B**–**E**), *n* = 3 (**F**)). A Kruskal Wallis test with Dunn’s multiple comparisons test was performed. Results with *p* < 0.05 were determined to be significant (* *p* < 0.05).

**Table 1 ijms-23-07384-t001:** Transcriptomics analysis. Selected statistically significant increased and decreased transcripts (FDR < 0.05).

Gene Symbol	Ensembl No. (Transcript)	log2-Fold (hEVs/nEVs)	*p* Value (FDR)
CTSD	ENST00000236671.7	29.37	<0.0001
HSP90B1	ENST00000299767.10	28.45	<0.0001
IGFBP6	ENST00000650247	27.55	<0.0001
MMP2	ENST00000543485	27.53	<0.0001
BNIP2	ENST00000415213	27.15	<0.0001
BCL2L12	ENST00000246784	26.53	<0.0001
CFL1	ENST00000526975	26.51	<0.0001
TMBIM4	ENST00000398033	25.89	<0.0001
CCL18	ENST00000616054.2	25.17	<0.0001
PTGES	ENST00000340607	14.07	0.0001
VEGFB	ENST00000543462.1	13.11	0.0006
IGFBP4	ENST00000269593.5	10.62	0.0004
CCL2	ENST00000225831	−8.07	0.0495
NAP1L4	ENST00000632437.1	−8.35	0.0473
IDO1	ENST00000518237	−8.73	0.0367

**Table 2 ijms-23-07384-t002:** miR analysis. Statistically significant increased and decreased small RNAs (FDR < 0.1 and fold change ≥2 or ≤−2 were considered differentially expressed (shown as log2-fold)).

Gene Symbol	Accession	log2-Fold (hEVs/nEVs)	*p* Value (FDR)
hsa-piR-019914	DQ597347	4.27	<0.0001
hsa-piR-13624	DQ588594	4.15	<0.0001
hsa-piR-020829	DQ598677	3.99	<0.0001
hsa-pir-018570	DQ595536	3.83	<0.0001
hsa-pir-020541	DQ598252	3.61	0.0032
hsa-pir-001312	DQ571813	3.58	<0.0001
hsa-pir-014923	DQ590404	2.77	0.0095
hsa-pir-004153	DQ575660	2.57	0.0095
hsa-piR-016658	DQ592931	1.83	0.016
hsa-piR-020326	DQ597916	1.79	0.054
hsa-piR-000765	DQ570956	1.34	0.018
hsa-miR-125b-5p	MIMAT0000423	−1.27	0.040
hsa-let-7i-5p	MIMAT0000415	−1.37	0.098
hsa-miR-26a-5p	MIMAT0000082	−1.40	0.065
hsa-let-7a-5p	MIMAT0000062	−1.55	0.031
hsa-miR-125a-5p	MIMAT0000443	−1.80	0.015
hsa-let-7f-5p	MIMAT0000067	−1.92	0.010
hsa-let-7e-5p	MIMAT0000066	−2.34	0.047

**Table 3 ijms-23-07384-t003:** Network analysis of all seven significantly down-regulated miRs (miR and genes). Analysis was done with MIENTURNET and miRTarBase, setting “strong evidence categories”, 0.05 as a threshold for the adjusted *p*-value (FDR). Transcripts from genes in red were found to be significantly upregulated after hypoxic pretreatment (Appendix A).

hsa-miR-125b-5p					hsa-let-7a-5p			hsa-miR-125a-5p		hsa-miR-26a-5p		hsa-let-7e-5p	hsa-let-7f-5p	hsa-let-7i-5p
BMPR1B	PRDM1	IL6R	ICAM2	SFRP5	MYC	LIN28A	AURKB	CDKN1A	ESRRA	HMGA2	TET2	HMGA2	KLK6	SOCS1
EIF4EBP1	IRF4	ABTB1	SET	LIFR	NKIRAS2	CASP9	RAB40C	LIN28A	VPS51	HMGA1	PTPN13	EIF3J	PRDM1	IL13
HMGA2	TP53INP1	HK2	CCNJ	Fas	NF2	IL6	ARG2	NTRK3	SIRT7	CDK8	PIK3C2A	SMC1A	IL13	COPS8
HMGA1	STAT3	E2F2	ENPEP	IGF1R	NRAS	E2F2	TNFRSF10B	CD34	CLEC5A	EZH2	TDG	WNT1	MPL	COPS6
GLI1	IGF2	MMP13	CSNK2A1	NEU1	PRDM1	IGF2BP1	TGFBR3	ERBB3	GALNT14	SMAD4	PHB	CCND1	CYP19A1	GPS1
NKIRAS2	LIN28B	MAPK14	MEGF9	ALOX5	TRIM71	MPL	LIN28B	ERBB2	STAT3	MYC	NRAS	MPL	COPS8	BMP4
SMO	PPP1CA	EPO	MAN1B1	EGFR	NR1I2	CCR7	PARP1	BAK1	MCL1	GDAP1	E2F2	AGO1	GPS1	IGF1
VDR	PRKRA	MUC1	AHRR	ANGPT2	VDR	CDKN1A	WNT1	ARID3B	MAPK14	CCNE1	PIK3CG	IGF1R	CCND1	AGO1
BAK1	BCL2	NES	SCNN1A	FES	HMGA2	EGFR	STAT3	TNFAIP3	EDN1	NOS2	IGF1	IGF1	COPS6	NEUROG1
ERBB3	TNFAIP3	CEBPA	VPS51	CDKN2D	HMGA1	RRM2	MAP4K4	SMAD4	PIK3CG	IL6	PLOD2	LIN28A	DYRK2	IL2
ERBB2	PIGF	ARID3A	SIRT7	JAK2	UHRF2	AGO1	TNFAIP3	MMP11	AKT1	MCL1	FUT8	AURKB	CCL7	AURKB
BMF	TBC1D1	MXD1	TET2	PTH1R	IGF2	EZH2		HK2	MEG3	AMACR	ST3GAL6	EZH2	AGO1	
NTRK3	FGFR2	PIAS3	SEMA4C	HOTTIP				EIF4EBP1	HDAC5	HGF	TRPC6	TNFRSF10B	IL6	
LIN28A	ARID3B	PCTP	SPHK1	DRAM2				LIFR	RAF1	LIN28B	LARP1	ARID3A	POSTN	
AKT1	SMAD4	GSS	MMP26					EGFR	JAK2	ZCCHC11		TNFAIP3		
RAF1	MCL1	IKZF3	MAP3K11					BCL2	ARID3A					

## Data Availability

Raw data are available as Appendix A.

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
