# Peer review of "Effects of Hypoxia on RNA Cargo in Extracellular Vesicles from Human Adipose-Derived Stromal/Stem Cells"

_ijms, 2022, doi:10.3390/ijms23137384_

Round 1

Reviewer 1 Report

The paper is interesting; however, it requires a few improvements prior to publishing. 

Here are more details questions and comments:

1. How many cell culture tests were performed to isolate mRNA? How many repetitions of the cell tests were performed?

2. The quality of figures has to be improved as some of them are difficult to read.

3. Fonts in Figure 3 are different

4. In the manuscript you refer to Figure 7 before referring to Figure 6. Is the sequence of figures correct in the paper?

5. Do you refer to Figures 6 A and 6B? Is Figure 6 A necessary? It is not clear what it shows.

Author Response

REVIEWER 1

The paper is interesting; however, it requires a few improvements prior to publishing. 

Answer: We would like to express our appreciation for the valuable comments. We have considered your comments carefully and have made the corrections suggested (marked in red in the revised document).

Here are more details questions and comments:

  1. How many cell culture tests were performed to isolate mRNA? How many repetitions of the cell tests were performed?

We used data from 4 independent biological replicates. Therefore, EVs from 4 different ASC preparations and also renal tubular epithelial cells from four different patients were used. We added this information to the figure legend of Fig. 7. PCR analysis was done in replicates from each sample.

  1. The quality of figures has to be improved as some of them are difficult to read.

We thank the reviewer for this advice. All figures are at high quality (300 dpi), but we think that this note is particularly aimed at Fig 6.

We have now changed and enlarged the labeling of the bars in Fig. 6 C and D. These graphs are thus easier to read.

  1. Fonts in Figure 3 are different

We thank the reviewer for this advice – we have now revised the font in Fig. 3.

  1. In the manuscript you refer to Figure 7 before referring to Figure 6. Is the sequence of figures correct in the paper?

This is correct – thank you very much. It was our mistake (page 10), we meant to refer to Fig. 6 A and B (and not to Fig. 7). We have now corrected this error.

  1. Do you refer to Figures 6 A and 6B? Is Figure 6 A necessary? It is not clear what it shows.

Correct –  this issue arose from our incorrect reference to Fig. 7 on page 10 (see answer to last concern). We have now corrected this.

In our opinion, Fig. 6 A is very helpful to the reader and visualizes the whole network of the seven down-regulated miRs. We have shown this network (Fig. 6A) to represent how extensively the miRs found interact with their potential targets/transcripts.

Reviewer 2 Report

This manuscript is of interest; however, there are several deficits in the experimental design about the effects of extracellular vesicles (EVs) on the pathophysiological functions of renal tubular epithelial cells in vitro and in vivo.

1. To increase the impact of this study, please experimentally address the effect of nEVs and hEVs on inflammatory response and fibrotic response upon the pathological stimuli in renal tubular epithelial cells.

2. In vivo models with AKI or CKD are highly recommended.

Author Response

REVIEWER 2

This manuscript is of interest; however, there are several deficits in the experimental design about the effects of extracellular vesicles (EVs) on the pathophysiological functions of renal tubular epithelial cells in vitro and in vivo.

Answer: We would like to express our appreciation for the valuable comments. We have considered your comments carefully and have made the corrections suggested (marked in red in the revised document).

  1. To increase the impact of this study, please experimentally address the effect of nEVs and hEVs on inflammatory response and fibrotic response upon the pathological stimuli in renal tubular epithelial cells.

We thank the reviewer for this valuable comment. The induction of cell injury (e.g. through a pathological/chemical stimulation) in cultured epithelial cells is a basis for the induction of regeneration / regenerative processes. Nevertheless, also isolation of epithelial cells for cell culture is based on destruction of epithelial integrity. The consequences are manifold: cell polarity and specific cell functions are lost; cells acquire non-epithelial characteristics and start to proliferate. This situation may also occur in situ when parts of the epithelium are lost, either by apoptosis or necrosis by organ or tissue injury. During recovery, surviving epithelial cells proliferate and restore the epithelial integrity and finally re-differentiate into highly differentiated epithelial cells. In vitro, this re-differentiation is mostly not complete due to sub-optimal culture conditions. Therefore, cultured epithelial cells resemble wounded or injured epithelia. For example, renal tubuluar epithelial cells used in our current study express vimentin which clearly indicates their injured character. The co-expression of cytokeratin and vimentin in cultured epithelial cells indicates the immature or wounded character of the cells (Gröne et al., 1987). The intermediate filament protein vimentin is normally expressed in mesenchymal cells and not present in differentiated epithelial cells. Nevertheless, in vitro vimentin expression did not decline after cells reached the state of confluence (Wallin et al., 1992), which shows that the re-differentiation of the cells has not been completed.

 Therefore, no additional pathological stimulation was done in the current study. We have now made this clearer in the text and added additional parts.

  1. In vivo models with AKI or CKD are highly recommended.

That is absolutely correct. Follow-up studies in an in vivo model should be performed and show the effects of EVs after hypoxic preconditioning. Nevertheless, the current study should first characterize only the EVs and show the different cargo between nEVs and hEVs, but also show effects on renal tubule epithelial cells in vitro.

 References:

Gröne HJ, Weber K, Gröne E, Helmchen U, Osborn M. Coexpression of keratin and vimentin in damaged and regenerating tubular epithelia of the kidney. Am J Pathol 1987;129:1–8.

Wallin A, Zhang G, Jones TW, Jaken S, Stevens JL. Mechanisms of the nephrogenic repair response. Lab Invest 1992;66:474–84.